# Clinical Efficacy of Ruxolitinib in Patients with Myelofibrosis: A Nationwide Population-Based Study in Korea

**DOI:** 10.3390/jcm10204774

**Published:** 2021-10-18

**Authors:** Byung-Hyun Lee, Hyemi Moon, Jae-Eun Chae, Ka-Won Kang, Byung-Soo Kim, Juneyoung Lee, Yong Park

**Affiliations:** 1Department of Internal Medicine, Korea University College of Medicine, Anam Hospital, Seoul 02841, Korea; potato0430@hanmail.net (B.-H.L.); ggm1018@gmail.com (K.-W.K.); kbs0309@korea.ac.kr (B.-S.K.); 2Department of Biostatistics, Korea University College of Medicine, Seoul 02841, Korea; mhm411@korea.ac.kr (H.M.); chaejaeeun@korea.ac.kr (J.-E.C.); jyleeuf@korea.ac.kr (J.L.)

**Keywords:** myelofibrosis, ruxolitinib, National Health Insurance, propensity score, prognosis

## Abstract

Previous studies have reported the survival benefit after ruxolitinib treatment in patients with myelofibrosis (MF). However, population-based data of its efficacy are limited. We analyzed the effects of ruxolitinib in MF patients with data from the Korean National Health Insurance Database. In total, 1199 patients diagnosed with MF from January 2011 to December 2017 were identified, of which 731 were included in this study. Patients who received ruxolitinib (*n* = 224) were matched with those who did not receive the drug (*n* = 507) using the 1:1 greedy algorithm. Propensity scores were formulated using five variables: age, sex, previous history of arterial/venous thrombosis, and red blood cell (RBC) or platelet (PLT) transfusion dependence at the time of diagnosis. Cox regression analysis for overall survival (OS) revealed that ruxolitinib treatment (hazard ratio (HR), 0.67; *p* = 0.017) was significantly related to superior survival. In the multivariable analysis for OS, older age (HR, 1.07; *p* < 0.001), male sex (HR, 1.94; *p* = 0.021), and RBC (HR, 3.72; *p* < 0.001) or PLT (HR, 9.58; *p* = 0.001) transfusion dependence were significantly associated with poor survival, although type of MF did not significantly affect survival. Considering evidence supporting these results remains weak, further studies on the efficacy of ruxolitinib in other populations are needed.

## 1. Introduction

Primary myelofibrosis (MF) is a subgroup of stem cell-derived clonal myeloproliferative neoplasms, characterized by clinical manifestations including severe anemia, thrombocytosis, bleeding, and splenomegaly [1]. MF that occurs following prior polycythemia vera (PV) or essential thrombocythemia (ET) is known as secondary MF [2]. According to the Korea National Cancer Incidence Database, the incidence rates of MF increased in the country from 0.08 per 100,000 person-years in 2003 to 0.15 per 100,000 person-years in 2011. The Health Insurance Review and Assessment Service database reported that the prevalence rates of MF also increased from 0.77 per 100,000 person-years in 2003 to 2.60 per 100,000 person-years in 2011 [3].

Ruxolitinib, a Janus kinase (JAK)1/JAK2 inhibitor, is the first U.S. Food and Drug Administration-approved drug for the treatment of intermediate- or high-risk MF, including primary MF, post-PV MF, and post-ET MF [4]. Until recently, the primary goal of therapy for patients with MF was alleviation of symptoms. A study reported that conventional therapies were unable to satisfactorily modify the biology of the disease [5]. Furthermore, the study also reported that ruxolitinib reduced the proinflammatory state and, consequently, alleviated the symptoms and decreased clonal myeloproliferation with possible leukemic transformation [5]. Two phase 3 clinical trials (COMFORT-I and COMFORT-II) demonstrated that ruxolitinib reduced spleen volume and improved the symptoms and quality of life of patients with intermediate-2 or high-risk MF [6,7]. The 3-year and 5-year follow-up results of the COMFORT-I and -II trials demonstrated a survival benefit among patients who received ruxolitinib compared to those treated with placebo and the best available therapy, respectively [8,9,10]. Exploratory analysis of the 5-year data pooled from these phase 3 trials revealed that the survival benefit with ruxolitinib was observed irrespective of baseline anemia status or transfusion requirements [11]. Although a survival benefit of ruxolitinib has been reported in several previous studies, other studies stated that the evidence was insufficient to conclude regarding the efficacy of the drug in MF [12,13]. Thus, the effect of ruxolitinib on the survival of patients remains controversial.

The present study aimed to evaluate efficacy of ruxolitinib based on real-world population data from the Korean National Health Insurance (NHI) Database (NHID). We also analyzed the effect of ruxolitinib based on progression to leukemia, and occurrence of thrombotic diseases. In addition, we compared ruxolitinib efficacy between patients with primary and secondary MF. In Korea, the National Health Insurance Service (NHIS) has provided financial support to patients with rare incurable diseases by reducing the medical expenses, including those of medications and transfusions. All medical institutions enter into a contract with the national government under a mandatory system, wherein all prescriptions, orders, and diagnostic codes are computerized and recorded in the NHID. Given this, the NHID provides data that are representative of the entire population than individual, smaller samples, which enabled us to conduct this population-based study.

## 2. Materials and Methods

### 2.1. Data Source

As a public organization, the NHIS provides health insurance to all citizens living in Korea and takes the responsibility for functioning of the system. The NHI covers 97% of the population, while the remaining 3% are covered by the Medical Aid program. The NHIS, as the single insurer, pays costs based on the billing records of the health care providers. The NHIS built a data warehouse, the NHID, to collect the required information on insurance eligibility, insurance contributions, medical history, and medical institutions. The NHID collects nationwide medical information, such as details of inpatient and outpatient diagnoses, treatment procedures, drug prescriptions, treatment duration, and costs.

Data on patients with MF for this study were obtained from the NHID. We collected information related to study outcomes including death, occurrence of leukemia, and thrombotic complications, and the number of transfusions of all patients, except those who were lost to follow-up. However, detailed information on individual patients was unavailable, including data regarding dose of medications or laboratory investigations. The diagnoses/definitions of diseases were based on the 10th revision of the International Statistical Classification of Diseases and Related Health Problems (ICD-10), as mentioned in the discharge reports of the patients and/or more than twice in their outpatient records. The disease codes used in this study were identical to those in the ICD-10. The Institutional Review Board of Korea University Medical Center approved this study. The requirement for informed consent was waived for the collection and analysis of data from the NHID.

### 2.2. Patient Selection

Data of patients with any medical visit encoded as MF (ICD-10 code D47.4) between 1 January 2011, and 31 December 2017, were extracted from the NHID. The first ICD-10-based disease code was implemented on 1 January 2011. The date of the first reported D47.4 code was considered the index date. Initially, we set a 6-month washout period to exclude prevalent MF and include only newly diagnosed MF patients for analysis during the study period. Thus, patients with an index date between 1 January 2011, and 30 June 2011, were excluded and those with initial disease code registered after 1 January 2011, were considered newly diagnosed MF. We estimated the washout period as 6 months because patients with MF would visit the clinic at least once in six months. Second, to eliminate confounding factors/confounders, such as primary hematologic malignancies, which are expected to have poor prognosis, patients who were diagnosed with leukemia (ICD-10 codes C91–95), myelodysplastic syndromes (ICD-10 codes D46), multiple myeloma (ICD-10 codes C90), and lymphomas (ICD-10 codes C81–86 and C88) before MF were excluded. Finally, patients younger than 18 years of age were excluded.

Patients with MF included in this study were divided into two groups based on ruxolitinib treatment. Patients who received ruxolitinib were matched with those who did not receive the drug using the 1:1 greedy algorithm, within a caliper width of 0.2 of the standard deviation of the logit of the estimated propensity score. Propensity scores were calculated using five variables: age, sex, previous history of thrombotic diseases, and red blood cell (RBC) or platelet (PLT) transfusion dependence. Patients in the two groups were matched based on the index dates that were less than a year apart. A previous history of thrombotic diseases was defined as the occurrence of ICD-10 codes starting with I20–22 (angina and myocardial infarction), I24–26 (ischemic heart disease), I63 (cerebral infarction), I70 (atherosclerosis), I74 (arterial embolism and thrombosis), I81 (portal vein thrombosis), I82 (other venous embolism and thrombosis), K64 (stroke), or I51.3 (intracardiac thrombosis) before the index date. Transfusion dependence was calculated as receipt of more than 8 units of RBC or 32 units of PLT (4 units as apheresis PLT) within 16 weeks from the index date based on the criteria described in a previous report [14].

### 2.3. Definitions

Secondary MF was defined as MF with prior diagnosis of PV (ICD-10 code D45), ET (ICD-10 code D47.3), and chronic myeloproliferative disease (ICD-10 code D47.1): post-PV and post-ET MF including MF after unspecified chronic myeloproliferative disease. Primary MF was defined as the condition that was not included in the diagnostic criteria of secondary MF. In this study, the primary outcome was overall survival (OS), defined as the time from the index date to death due to any cause or the end of the study (31 December 2017), whichever came first. Patients who underwent allogeneic stem cell transplantation were censored at the time of transplantation. Secondary outcomes were occurrence of leukemia (excluding chronic myelogenous leukemia [ICD-10 codes C92.1 and C92.2] and lymphoid leukemia [ICD-10 code C91]) and thrombotic diseases, which were newly identified after the index date.

### 2.4. Statistical Analysis

Patients who received ruxolitinib and those who did not were matched according to the propensity score. The propensity score was estimated using a logistic regression model. Baseline characteristics of matched pairs were compared using the paired t-test for continuous variables and McNemar’s test for categorical variables. The standardized difference was also used to compare the balance in baseline characteristics. Survival curves were estimated using the Kaplan–Meier method and differences in the survival distributions were evaluated using the log-rank test. Considering matched data, stratified multiple Cox proportional hazard regression analyses were used to estimate the effects of ruxolitinib treatment on the outcomes, and subgroup analyses were conducted to compare the effects of ruxolitinib treatment between patients with primary and secondary MF. Interaction examinations between the treatment and type of MF were performed using the Cox regression model. Further subgroup analyses were performed to evaluate the associations between the outcomes including OS, occurrence of leukemia or thrombotic diseases and prognostic factors in the patients who underwent treatment with ruxolitinib using forest plots. Hazard ratios (HRs) and their 95% confidence intervals (CIs) were calculated. All tests were two-sided, and *p*-values of <0.05 were considered significant. Statistical analyses were performed using SAS software, version 9.4 (SAS Institute, Cary, NC, USA).

## 3. Results

### 3.1. Study Polulation

In total, 1199 patients with MF were identified between 1 January 2011, and 31 December 2017. After excluding patients with newly diagnosed MF within the washout period (*n* = 117), patients with previously diagnosed hematologic malignancies (*n* = 341), leukemia (*n* = 227, including 112 patients with chronic myelogenous leukemia), myelodysplastic syndromes (*n* = 68), multiple myeloma (*n* = 18), and lymphomas (*n* = 28) were excluded. Subsequently, patients younger than 18 years (*n* = 10) of age were excluded. Finally, 731 patients with MF were included in this study (Figure 1). Of the 731 patients, 224 (30.6%) received ruxolitinib and 507 (69.4%) did not receive the drug. The baseline characteristics of the patients who received ruxolitinib were significantly different from those who did not receive the drug before matching (Table 1). Patients who underwent treatment with ruxolitinib were significantly older than those who did not receive the drug (66.0 vs. 60.8 years; *p* < 0.001). The rate of previous thrombotic diseases was significantly higher among patients in the ruxolitinib group than among those in the non-ruxolitinib group (56.7% vs. 48.3%; *p* = 0.037). The rate of PLT transfusion dependence was significantly lower among patients in the ruxolitinib group than among those in the non-ruxolitinib group (0.45% vs. 8.88%; *p* < 0.001). No significant difference was observed based on sex (*p* = 0.889), rate of RBC transfusion dependence (*p* = 0.208), or rate of previous hydroxyurea treatment (*p* = 0.331). After matching, 224 patients each were selected from the ruxolitinib and non-ruxolitinib groups, and no significant differences were observed in the baseline variables between them (*p* = 0.192–1.000). Among the 731 patients with MF, 391 (53.5%) had primary MF and 340 (46.5%) had secondary MF. Among the 391 patients with primary MF, 100 (25.6%) patients received ruxolitinib and 291 (74.4%) did not receive the drug. Among the 340 patients with secondary MF, 124 (36.5%) patients received ruxolitinib and 216 (63.5%) did not receive the drug.

Incident and prevalent cases of MF during the study period from 2011 to 2017 were showed in Figure 2. Incident cases of MF gradually increased from 2011 (44 persons per year) to 2017 (165 persons per year). Among the patients with primary MF, incident cases continuously increased from 2011 (18 persons per year) to 2017 (76 persons per year). Among the secondary MF, incident cases of MF consistent from 2011 to 2014 (22 persons per year to 32 persons per year), and then gradually increased from 2015 (59 persons per year) to 2017 (89 persons per year). Prevalent cases of MF were also continuously increased from 2011 (44 persons) to 2017 (464 persons).

### 3.2. Clinical Outcomes Based on Ruxolitinib Treatment

Five hundred seven (69.4%) patients visited the clinic at least once a month, 152 (20.8%) visited twice a month, 42 (5.8%) visited thrice a month, and 30 (4.1%) patients visited more than thrice a month. On average, patients visited the clinic 1.13 times a month during the study period. The median OS of patients in the ruxolitinib group was longer than of those in the non-ruxolitinib group. The median OS of patients was 52 and 45 months in the ruxolitinib and non-ruxolitinib groups, respectively (*p* = 0.018; Figure 3). In the ruxolitinib and non-ruxolitinib groups, the 1-year, 2-year, and 4-year OS rates of the patients were 92% and 79%, 81% and 65%, and 57% and 47%, respectively. In the Cox regression analysis for OS, ruxolitinib treatment (HR, 0.67; 95% CI, 0.49–0.93; *p* = 0.017) was a significant prognostic factor for longer survival (Table 2). In the subgroup analysis, ruxolitinib treatment favorably affected survival in patients with primary MF (HR, 0.53; 95% CI, 0.35–0.81; *p* = 0.003). In contrast, no significant difference was observed in patients with secondary MF who underwent ruxolitinib treatment (HR, 0.91; 95% CI, 0.55–1.52; *p* = 0.729). Although, interaction between OS and type of MF (primary or secondary MF) was not significant (*p* = 0.104). Ruxolitinib treatment did not appear to influence the occurrence of leukemia and thrombotic complications in MF patients, and subgroup analyses in both primary and secondary MF also showed no significant differences (Appendix A). Then, to confirm the effect of ruxolitinib in survival prognosis, we conducted multivariable analyses including other relevant clinical factors in patients treated with ruxolitinib.

### 3.3. Subgroup Analysis in Patients Treated with Ruxolitinib

In patients treated with ruxolitinib, type of MF did not affect OS, with a crude HR of 1.22 (95% CI, 0.74–2.00; *p* = 0.434). In the multivariable Cox-regression analysis for OS, older age (adjusted HR, 1.07; *p* < 0.001), male sex (adjusted HR, 1.94; *p* = 0.021), and RBC (adjusted HR, 3.72; *p* < 0.001), PLT (adjusted HR, 9.58; *p* = 0.001) transfusion dependence were significantly associated with poor survival, although type of MF did not significantly affect survival (adjusted HR, 1.14; *p* = 0.656) (Figure 4A). In the analysis for secondary outcomes, RBC transfusion dependence was significant unfavorable risk factor for the occurrence of leukemia (adjusted HR, 2.24; *p* = 0.049) (Figure 4B). PLT transfusion dependence was significant risk factor for thrombotic complications (adjusted HR, 18.51; *p* = 0.002) (Figure 4C).

## 4. Discussion

In our nationwide population-based analysis, ruxolitinib treatment appeared to be associated with better survival in patients with MF. Previous phase 3 trials have reported the favorable effects of ruxolitinib in patients with MF [6,7]. The main study endpoint was the reduction in spleen volume in these studies, while OS was a secondary endpoint. Survival analysis was performed according to the intention-to-treat principle after a median follow-up of 51 weeks in the COMFORT-I trial and after 61.1 weeks in the COMFORT-II trial. No differences in survival between the two arms were observed in the COMFORT-II trial, whereas an advantage for the ruxolitinib arm was seen in the COMFORT-I trial [6,7]. Nonetheless, the effect of ruxolitinib on the survival of patients remains controversial because of several confounding factors. For example, patients enrolled in the COMFORT trials were followed up over an extended, non-controlled study phase, and the results were reported individually at 3 and 5 years by the two trials, while the combined results were reported at 5 years [8,9,10,11,14].

In the present study, we observed a favorable effect of ruxolitinib in patients with MF in the real-world population, although there were inevitable biases associated with the study design based on the data collected from the NHID. Although propensity score matching was conducted to correct biases between patients who received ruxolitinib and those who did not, patients with poor prognosis were more likely to receive the therapy. Nevertheless, our results showed favorable survival outcomes in ruxolitinib-treated patients with MF, who are expected to be included in higher risk categories than in those who did not receive the drug. Considering that the evidence supporting prolongation of survival by ruxolitinib therapy in patients with MF remains weak, the results of our population-based study might suggest a potential beneficial effect.

The prognostic implications for patients with primary and secondary MF are also controversial. One study did not report any morphologic, phenotypic, or prognostic differences between patients with primary and secondary MF [15], whereas another study reported that patients with post-ET MF demonstrated better OS than those with primary MF and post-PV MF [16]. The IPSS and dynamic IPSS are often applied to predict prognosis in patients with secondary MF; however, these models have been developed for primary MF patients, thus may not be optimal for the former group [17,18]. Likewise, there is insufficient information regarding the clinical efficacy of ruxolitinib in patients with primary and secondary MF. Although a previous study showed that patients who received ruxolitinib had a decrease in spleen volume and showed improvement in the total symptom score across MF subtypes, no significant differences were observed among patients with primary and secondary MF. Another retrospective study showed that ruxolitinib treatment resulted in reduction of splenomegaly and alleviated symptoms in patients with primary and secondary MF [19]. However, reports comparing survival outcomes based on MF types in patients who have undergone ruxolitinib treatment are limited. In this study, the survival benefit of ruxolitinib was not related to type of MF. Although ruxolitinib treatment significantly affected better survival in primary MF in Kaplan–Meier and univariable Cox analysis, difference between primary and secondary MF was not statistically significant. Moreover, in the subgroup of patients treated with ruxolitinib, multivariable analysis showed that type of MF was not a significant prognostic factor for survival of MF patients. Because we did not include the type of MF (primary or secondary) as a variable for propensity score, the fractions of patients with primary and secondary MF are not balanced in the otherwise matched groups of patients and might lead to conflicting results for type of MF on survival. These results suggests that the target population for ruxolitinib treatment needs to be evaluated.

The present study has several limitations. First, this study conducted on the basis of the public database. Even though enrolled patients’ data were initially matched using propensity score and no significant difference was observed in the baseline characteristics of patients treated with ruxolitinib or not, this is not a randomized study and presumed to have unevaluated factors affecting the study outcomes. Second, the NHID provided limited information regarding diagnosis, drugs, and service codes. Detailed information of individual patients were unavailable, including dose of medication; laboratory data such as hemoglobin level, white blood cell or PLT counts, percentage of circulating blasts; and the performance status or constitutional symptoms of patients. In addition the prognostic relevance of cytogenetics was highlighted in a recent study [20]. However, we were unable to perform cytogenetic analysis in this study. Third, we used the ICD-10 code to define secondary MF including post-PV, post-ET, and post-chronic myeloproliferative disease. Thus, we would have overestimated patients with secondary MF because categories of chronic myeloproliferative disease were broad and ambiguous. Fourth, we used the Kaplan–Meier method and the log-rank test to compare the survival distributions of ruxolitinib and no ruxolitinib groups. However, Kaplan–Meier curves cross at year 5. If there are large number of events, and events in one group tend to occur later than in other group on average, test can obtain significant result even if survival curves cross. Although we could conclude that patients in ruxolitinib group died significantly later than in no ruxolitinib group, alternative statistical tests can be needed to validate the results. Finally, all patients included in this study were Korean. Applying our results to other populations was cautiously considered.

In conclusion, this study shows that ruxolitinib treatment could be effective in patients with MF based on real-world population based database. In addition, older age, male sex, and RBC or PLT transfusion dependence were significant prognostic factors for poor survival in patients who received ruxolitinib, although type of MF was not. Further studies are needed to evaluate the efficacy of ruxolitinib in other populations.

## Figures and Tables

**Figure 1 jcm-10-04774-f001:**
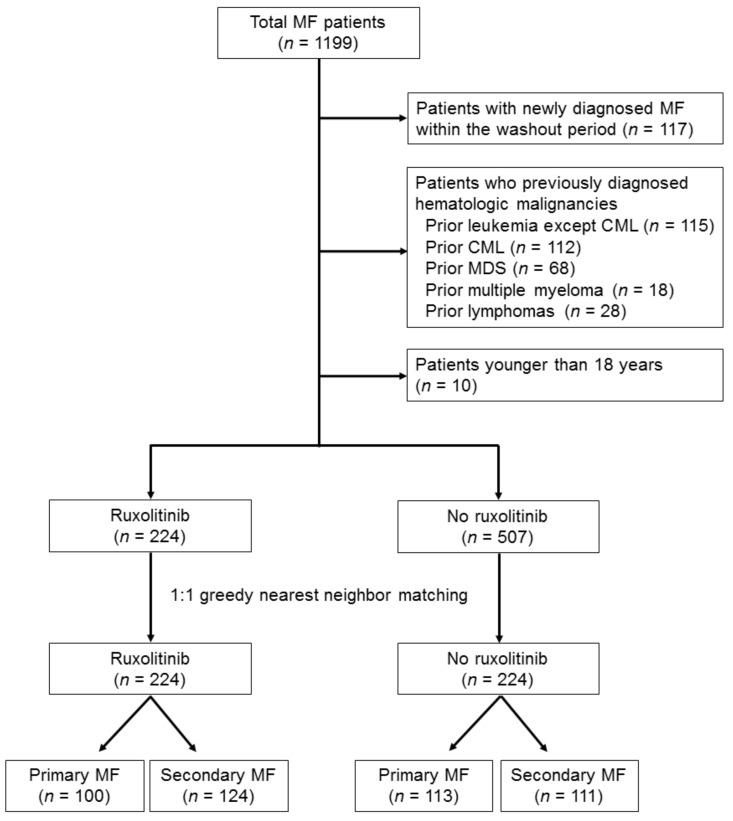
Flow diagram of the 1199 patients with myelofibrosis included in this study.

**Figure 2 jcm-10-04774-f002:**
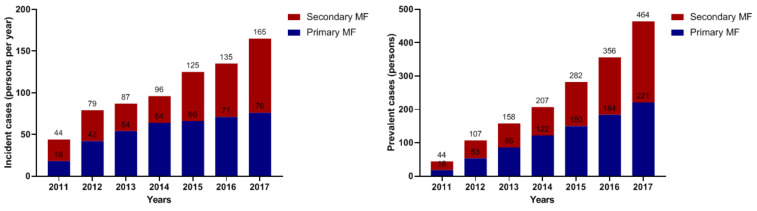
Incident and prevalent cases of myelofibrosis during the study period from 2011 to 2017.

**Figure 3 jcm-10-04774-f003:**
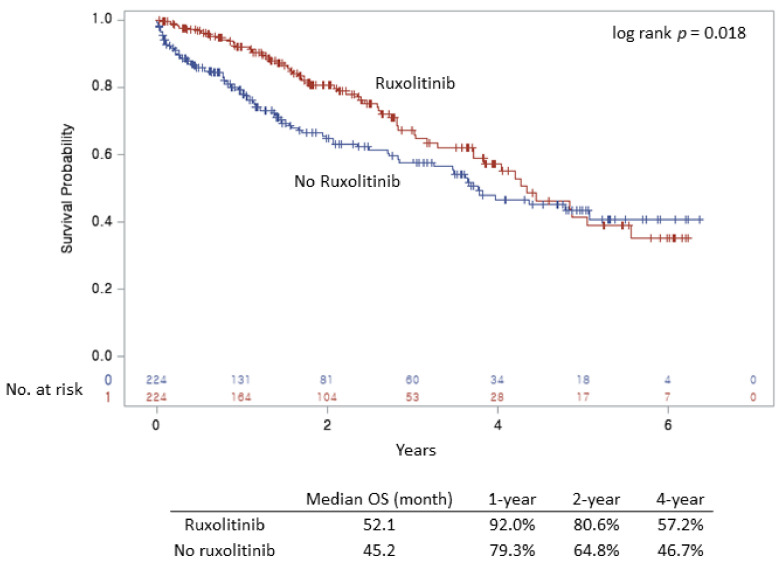
Kaplan–Meier curves for overall survival in myelofibrosis patients.

**Figure 4 jcm-10-04774-f004:**
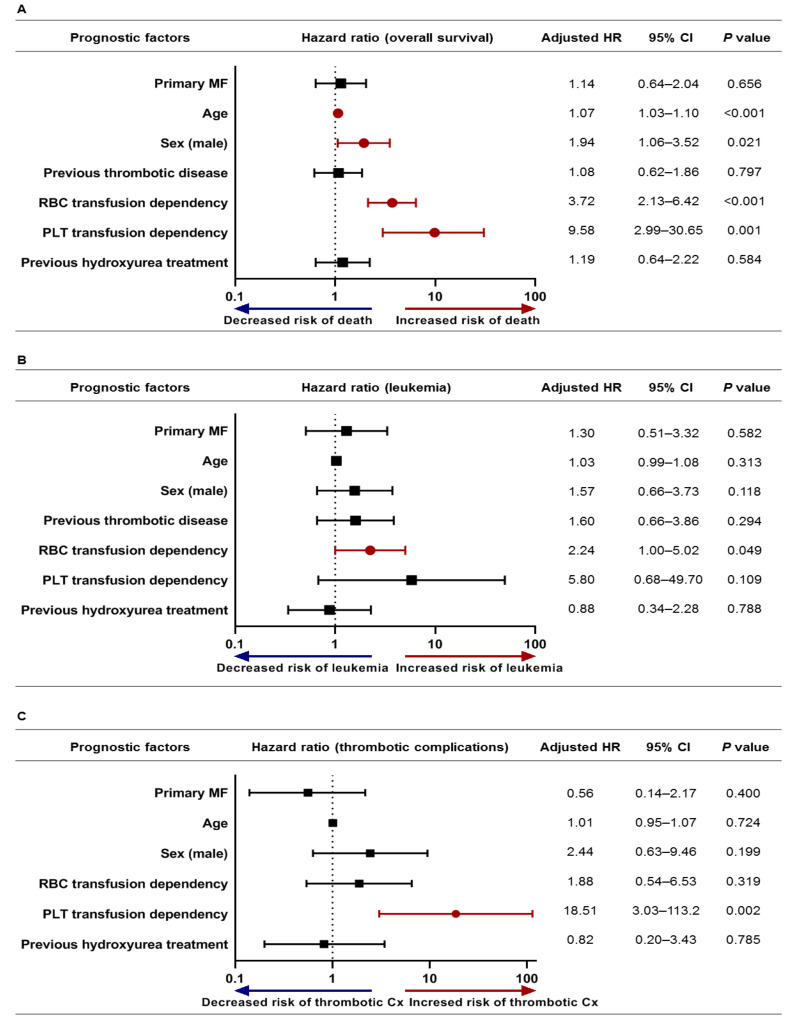
Multivariable Cox regression analysis in myelofibrosis patients treated with ruxolitinib for (**A**) overall survival, (**B**) occurrence of leukemia, (**C**) thrombotic complications. HR, hazard ratio; PLT, platelet; RBC, red blood cell.

**Table 1 jcm-10-04774-t001:** Patient characteristics before and after propensity score matching.

	before Matching (*n* = 731)			after Matching (*n* = 426)		
	No Ruxolitinib (*n* = 509)	Ruxolitinib (*n* = 224)	*p*	No Ruxolitinib (*n* = 224)	Ruxolitinib (*n* = 224)	*p*
Age, years			<0.001 ^a^			0.192 ^c^
Mean (SD)	60.8 (16.1)	66.0 (11.1)		66.9 (12.3)	66.0 (11.1)	
Median (min, max)	63.0 (18.0, 92.0)	68.0 (28.0, 88.0)		68.5 (25.0, 92.0)	68.0 (28.0, 88.0)	
Sex, *n* (%)			0.889 ^b^			0.729 ^d^
Male	288 (56.8)	126 (56.3)		123 (54.9)	126 (56.3)	
Female	219 (43.2)	98 (43.7)		101 (45.1)	98 (43.7)	
Thrombotic complications, *n* (%)			0.037 ^b^			0.670 ^d^
Yes	245 (48.3)	127 (56.7)		131 (58.5)	127 (56.7)	
No	262 (51.7)	97 (43.3)		93 (41.5)	97 (43.3)	
RBC transfusion dependency, *n* (%)			0.208 ^b^			1.000 ^d^
Yes	121 (23.9)	44 (19.6)		44 (19.6)	44 (19.6)	
No	386 (76.1)	180 (80.4)		180 (80.4)	180 (80.4)	
PLT transfusion dependency, *n* (%)			<0.001 ^b^			1.000 ^d^
Yes	45 (8.9)	1 (0.5)		1 (0.5)	1 (0.5)	
No	462 (91.1)	223 (99.5)		223 (99.5)	223 (99.5)	
Prior hydroxyurea treatment, *n* (%)			<0.001 ^b^			0.251 ^d^
Yes	131 (25.8)	110 (49.1)		105 (46.9)	110 (49.1)	
No	376 (74.2)	114 (50.9)		119 (53.1)	114 (50.9)	

PLT, platelet; RBC, red blood cell; SD, standard deviation. ^a^
*p*-value by student’s *t*-test; ^b^
*p*-value by chi-square test; ^c^ *p*-value by paired *t*-test; ^d^ *p*-value by McNemar’s test.

**Table 2 jcm-10-04774-t002:** Stratified Cox regression analysis for OS in unmatched and matched cohorts.

	Unmatched						Propensity Score Matched					
	*n*	No. of Events	HR	95% CI	*p*	*p* for Interaction	*n*	No. of Events	HR	95% CI	*p*	*p* for Interaction
Overall												
No ruxolitinib	507	165	1.00				224	83	1.00			
Ruxolitinib	224	63	0.75	0.56–1.01	0.058		224	63	0.67	0.49–0.93	0.017	
						0.881						0.104
Primary MF												
No ruxolitinib	291	99	1.00				113	52	1.00			
Ruxolitinib	100	32	0.78	0.54–1.14	0.198		100	32	0.53	0.35–0.81	0.003	
Secondary MF												
No ruxolitinib	216	66	1.00				111	31	1.00			
Ruxolitinib	124	31	0.75	0.50–1.12	0.169		124	31	0.91	0.55–1.52	0.729	

CI, confidence interval; HR, hazard ratio; MF, myelofibrosis; OS, overall survival.

## Data Availability

Data are available on reasonable request to the corresponding author.

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
