# Peer review of "Clinical Efficacy of Ruxolitinib in Patients with Myelofibrosis: A Nationwide Population-Based Study in Korea"

_jcm, 2021, doi:10.3390/jcm10204774_

Round 1
Reviewer 1 Report
- Introduction line 51-57: This paragraph is not written clearly (content is not understandable to me).
- Results: Line 180: “incidence and prevalence in the study population”: incidence and prevalence should be referred to the whole population of a country or in this case maybe the population of the NHI. This paragraph should be clarified.
- Clinical outcomes based on Ruxo-Treatment: Line 194ff: is this the average visits
of patients in the clinic? Its maybe more interesting how many visits in average the patients had in the clinic during the study period
- Figure 3: Units of 1-4 year survival should be mentioned
- Figure 4: the caption below the Plots should be optimized: “favors no prognostic factor/ favors prognostic factor sounds kind of awkward in this context
- Figure 4: the usage of plot-diagrams and also the Kaplan- M plot should be mentioned in the statistical part.
- Discussion: regarding the differences between patients with pMF and sMF: by analyzing the subgroup of patients with pMF, you could show an influence of Ruxo Tx on survival, but when you look into the subgroup of Ruxo treated patients, than you get no affect of types of MF on survival.
Since you did not include the type of MF (primary or secondary) as an variable for propensity score, the fractions of patients with pMF and sMF are not balanced in the otherwise matched groups of patients (+ 25% of sMF in the Ruxo group). This might maybe explain the conflicting results and could be discussed
Author Response
Point 1: Introduction line 51-57: This paragraph is not written clearly (content is not understandable to me).
Response 1: Thank you for your valuable comment. We have modified this paragraph to make clear description.
Point 2: Results: Line 180: “incidence and prevalence in the study population”: incidence and prevalence should be referred to the whole population of a country or in this case maybe the population of the NHI. This paragraph should be clarified.
Response 2: As you pointed out, the study population is those of the NHI. Considering we evaluated the number of patients during the study period, it would be more appropriate to use the terms of “incident cases and prevalent cases” instead of “incidence and prevalence.” We have modified this paragraph and Figure 2 legend accordingly. We hope that these revisions are acceptable.
Point 3: Clinical outcomes based on Ruxo-Treatment: Line 194ff: is this the average visits of patients in the clinic? Its maybe more interesting how many visits in average the patients had in the clinic during the study period
Response 3: This sentence is about the average number of clinic visits per a month. As you recommended, we have added information that how many visits in average the patients had in the clinic to the Clinical Outcomes Based on Ruxolitinib Treatment subsection in the results section.
Point 4: Figure 3: Units of 1-4 year survival should be mentioned
Response 4: This has been done.
Point 5: Figure 4: the caption below the Plots should be optimized: “favors no prognostic factor/ favors prognostic factor sounds kind of awkward in this context
Response 5: We agree with your opinion. We modified the caption accordingly.
Point 6: Figure 4: the usage of plot-diagrams and also the Kaplan- M plot should be mentioned in the statistical part.
Response 6: We have included this information in the method section.
Point 7: Discussion: regarding the differences between patients with pMF and sMF: by analyzing the subgroup of patients with pMF, you could show an influence of Ruxo Tx on survival, but when you look into the subgroup of Ruxo treated patients, than you get no affect of types of MF on survival.
Since you did not include the type of MF (primary or secondary) as an variable for propensity score, the fractions of patients with pMF and sMF are not balanced in the otherwise matched groups of patients (+ 25% of sMF in the Ruxo group). This might maybe explain the conflicting results and could be discussed
Response 7: Thank you very much for your valuable comment. We have incorporated your comment and additional explanation to the discussion section. Thank you again for this insightful suggestion and help to improve our work.
Reviewer 2 Report
Abstract:
- Introduce acronyms for overall survival (OS), red blood cells (RBC) and platelets (PLT) in the abstract
- The abstract conclusion could be strengthened by providing reasons why more studies are needed to evaluate the efficacy of ruxolitinib in MF
Methods:
- Please indicate the statistical test used for OS (ruxolitinib vs. no ruxolitinib)
Table 1:
- Please indicate statistical tests used to determine P in the table legend
- The number of patients who received PLT transfusion in the no ruxolitinib group after matching does not add up to n=224 (45+462=507); please check and correct data
Figure 3:
- Indicate the statistical test used for OS (ruxolitinib vs. no ruxolitinib) in the figure legend
Figure 4:
- Explain abbreviations (RBC, PLT, HR) in the figure legend
- Please change language used in X axis legend (“Favors no prognostic factor” falsely implies that a high HR is not a prognostic factor)
Results and Discussion:
- It is crucial to address the fact that the Kaplan Meier curves (ruxolitinib vs. no ruxolitinib) cross at year 5. It will be necessary to state which statistical test was used, why this test was chosen and if/how the cross over influences the validity of the test
- Overall, the study would be stronger if MF grade and spleen size would be included in the study
Author Response
Point 1: Abstract:
Introduce acronyms for overall survival (OS), red blood cells (RBC) and platelets (PLT) in the abstract
Response 1-1: This has been done.
The abstract conclusion could be strengthened by providing reasons why more studies are needed to evaluate the efficacy of ruxolitinib in MF
Response 1-2: Thank you for your valuable comment. We have included this information at the end of the abstract.
Point 2: Methods:
Please indicate the statistical test used for OS (ruxolitinib vs. no ruxolitinib)
Response 2: We have included this information in the method section.
Point 3: Table 1:
Please indicate statistical tests used to determine P in the table legend
Response 3-1: This has been done.
The number of patients who received PLT transfusion in the no ruxolitinib group after matching does not add up to n=224 (45+462=507); please check and correct data
Response 3-2: This has been corrected.
Point 4: Figure 3:
Indicate the statistical test used for OS (ruxolitinib vs. no ruxolitinib) in the figure legend
Response 4: This has been done.
Point 5: Figure 4:
Explain abbreviations (RBC, PLT, HR) in the figure legend
Response 5-1: This has been done.
Please change language used in X axis legend (“Favors no prognostic factor” falsely implies that a high HR is not a prognostic factor)
Response 5-2: We have modified this accordingly.
Point 6: Results and Discussion:
It is crucial to address the fact that the Kaplan Meier curves (ruxolitinib vs. no ruxolitinib) cross at year 5. It will be necessary to state which statistical test was used, why this test was chosen and if/how the cross over influences the validity of the test
Response 6-1: Thank you for your valuable comment. We used the log-rank test to compare the survival distributions of ruxolitinib and no ruxolitinib groups. We think that crossing of survival curves does not completely invalidate the log-rank test. If there are large number of events, and events in one group tend to occur later than in other group on average, test can obtain significant result even if survival curves cross. Although we think that we could conclude that patients in ruxolitinib group died significantly later than in no ruxolitinib, alternative statistical tests can be needed to validate the results. We have included this information as a limitation in the discussion section.
Overall, the study would be stronger if MF grade and spleen size would be included in the study
Response 6-2: We agree that MF grade and spleen size would be included in the study. However, the NHI database do not provide these information and detailed information of individual patients are unavailable. We apologize for this issue.